

# Potassium isotopic variability and implications for $^{40}$K-based geochronology

Leah E. Morgan[1]

[1]U.S. Geological Survey, Denver, CO 80225, USA

*Correspondence to*: Leah E. Morgan (lemorgan@usgs.gov)

**Abstract.** $^{40}$Ar/$^{39}$Ar and K-Ar geochronology assume that $^{40}$K/K values are invariant among the sample of interest, the co-irradiated neutron fluence monitor ("standard"), and the material used to measure decay constants. Until recently, this assumption was reasonable due to the small K isotope ($^{41}$K, $^{40}$K, $^{39}$K) variability found in many terrestrial samples and the

negligible effect of any variation relative to the precision of the determined age. The recent discovery of measurable $\delta^{41}$K variability in terrestrial samples now questions this assumption. Although $\delta^{41}$K values for some neutron fluence monitors have now been reported, potassium isotopes are not routinely measured on samples dated by the $^{40}$Ar/$^{39}$Ar method even though a wide range of silicate materials were found to vary by >2.5‰. Further, the $^{40}$K decay constants used in $^{40}$Ar/$^{39}$Ar geochronology are based on activity counting of radioactive decay in K-rich salts. These salts have not been measured for $\delta^{41}$K, yet evaporites

have been shown to vary by >1‰ from the mean value of silicates. The potential effects of $\delta^{41}$K variability on $^{40}$Ar/$^{39}$Ar ages are illustrated using the case of the ca. 28.2 Ma Fish Canyon sanidine (FCs) and the ca. 99 Ma Mt. Dromedary Biotite (GA-1550). If the two standards have $\delta^{41}$K values as measured and the material used to determine decay constants is appropriately represented by $\delta^{41}$K of evaporites, the age of FCs is underestimated by ca. 7 ka (0.25‰). Although this is a small effect, such bias is becoming important as the analytical precision and accuracy of isotopic measurements and calculation of $^{40}$Ar/$^{39}$Ar ages

continue to improve.

## 1 $^{40}$Ar/$^{39}$Ar geochronology

The $^{40}$Ar/$^{39}$Ar system is a variant of K-Ar geochronology, where $^{40}$K decays to $^{40}$Ar with a half-life of ca. 1.25 Ga (Merrihue and Turner, 1966). When applying the K-Ar method, potassium and argon concentrations are typically measured on separate aliquots of sample. Values for the branched decay constant of $^{40}$K to $^{40}$Ar and $^{40}$Ca (e.g. Steiger and Jäger, 1977) are combined

with the concentration measurements to calculate an age (Eq. (1)).

The $^{40}$Ar/$^{39}$Ar geochronology method requires unknown samples to be co-irradiated alongside neutron fluence monitors (often called "mineral standards") to convert some $^{39}$K to $^{39}$Ar (Dalrymple et al., 1981). By assuming a fixed $^{40}$K/$^{39}$K value, the $^{39}$Ar measured by mass spectrometry can be used as a proxy for the parent isotope $^{40}$K. Age calculations using the $^{40}$Ar/$^{39}$Ar method

rely on the measured $^{40}$Ar/$^{39}$Ar isotopic ratios in both sample and mineral standard, in addition to values for the total $^{40}$K decay





constant and the known age of the mineral standard. Often unrecognized, but integral to both the $^{40}Ar/^{39}Ar$ and K-Ar geochronology methods, is the assumption that potassium isotopic ratios (often known as $^{40}K/K$) are invariant between three materials: (1) samples, (2) mineral standards (including any primary standards that secondary standards are intercalibrated with), and (3) materials on which the decay constant was measured. While this assumption has previously been reasonably

founded in the consistency of terrestrial stable K isotopic ratios (Humayun and Clayton, 1995), more recent work (e.g. Li et al., 2016; Morgan et al., 2018; Ramos et al., 2018; Wang and Jacobsen, 2016) has identified terrestrial stable K isotopic variability and underscores the need to revisit the issue.

## 2 Potassium isotope measurements and variability

High-precision measurements of stable K isotopes ($^{39}K$, $^{41}K$) by multi-collector inductively coupled plasma mass spectrometry
(MC-ICP-MS) present a number of analytical challenges because the Ar carrier gas produces hydrides with isobaric interferences on both $^{39}K$ ($^{38}ArH^+$) and $^{41}K$ ($^{40}ArH^+$). Morgan et al. (2018) presented new high-precision measurements of $\delta^{41}K$ values (defined as $\delta^{41}K = 1000 \times (^{41}K/^{39}K_{sample} - {^{41}K/^{39}K_{standard}})/{^{41}K/^{39}K_{standard}}$, relative to NIST KCl elemental SRM 999b) on a large number (n = 85) of natural samples using a MC-ICP-MS (Thermo Scientific NEPTUNE Plus) instrument run in high-resolution mode and cold plasma conditions to reduce isobaric interferences.


A summary of the measured $\delta^{41}K$ values for all geological and biological samples and relevant supplemental data are shown in Fig. 1, modified from Morgan et al. (2018). The total variability observed in all samples ranges from -1.36‰ (N=2) in a lepidolite from the Glenbuchat pegmatite to 1.21‰ (N=5) in a kaliophilite from the Alban Hills, Italy. The average of n = 78 samples of bulk rock and mineral separates is -0.46 ± 0.80‰ (2σ). Silicate bulk rocks and minerals occupy the lower end of
this range, whereas at the heavy end of the range is seawater ($\delta^{41}K$ = 0‰), K-evaporite minerals (average $\delta^{41}K$ = +0.07 ± 0.69‰; 2σ, n =7), and two fertilizers ($\delta^{41}K$ = 0.12‰ and 0.03‰). Most silicate rocks and minerals fall between -0.7‰ and -0‰, while most evaporites fall between -0.2‰ and 0.2‰.

One small but important effect of these results is the need to revisit the atomic weight of K, which is determined by the
International Union of Pure and Applied Chemistry. Morgan et al. (2018) suggested the need to update this value for bulk earth from 39.0983 to 39.0982. This change will not appreciably affect most geochronology samples, but the ability to account for variable atomic weight is provided in the equations below.

## 3 Calculating the effects of potassium isotope variability

Determining the effects of potassium isotopic variability requires an understanding of the parameters involved in calculating
an $^{40}Ar/^{39}Ar$ age. The method ultimately relies on the K-Ar age equation:



$$t_{std} = \frac{1}{\lambda} \ln \left[ 1 + \frac{\lambda}{\lambda_e} \frac{{}^{40}Ar^{*}{}_{std}}{{}^{40}K_{std}} \right] \qquad (1)$$

where $t_{std}$ is the K-Ar age of a standard, $\lambda$ and $\lambda_e$ are the total and electron capture branches of the ${}^{40}K$ decay constant, ${}^{40}Ar^{*}{}_{std}$ is the abundance of radiogenic ${}^{40}Ar$ in the standard, and ${}^{40}K_{std}$ is the abundance of ${}^{40}K$ in the standard. Values for calculating ${}^{40}K_{std}$ can be substituted in Eq. (1):

$$t_{std} = \frac{1}{\lambda} \ln \left[ 1 + \frac{\lambda}{\lambda_e} \frac{{}^{40}Ar^{*}{}_{std} \times atwtK}{\omega_K \times f_G} \right] \qquad (2)$$

where atwtK = atomic weight of K, $\omega_K$ = weight fraction of K in the neutron fluence monitor, and $f = {}^{40}K/K$ (previously
assumed to be constant, subscript 'G' indicates value from Garner et al. (1975)).

The equation for $\lambda$ from Min et al. (2000) is

$$\lambda = A \frac{atwtK \times Y}{f_G \times N_o} \qquad (3)$$

where A = activity of ${}^{40}K$, $N_o$ = Avogadro's Number, and Y = number of seconds in the mean solar year. Recalculating $\lambda$ using
values for atwtK and $f$ specific to the material in question yields

$$\lambda_{new} = \lambda \frac{f_G}{f_{new}} \frac{atwtK_{new}}{atwtK} \qquad (4)$$

where $\lambda_{new}$, $f_{new,}$ and $atwtK_{new}$ can be represented by $\lambda_\lambda$, $f_\lambda$, and $atwtK_\lambda$ for values of material used in decay constant determinations or $f_{std}$ and $atwtK_{std}$ for values of neutron fluence monitors.

The above equations can then be used to update the age of a neutron fluence monitor via the K-Ar equation to reflect new $\lambda$ and $f$ values:

$$t_{std} = \frac{1}{\lambda \frac{f_G}{f_\lambda} \frac{atwtK_\lambda}{atwtK}} \ln \left[ 1 + \frac{\lambda \frac{f_G}{f_\lambda} \frac{atwtK_\lambda}{atwtK}}{\lambda_e \frac{f_G}{f_\lambda} \frac{atwtK_\lambda}{atwtK}} \frac{{}^{40}Ar^{*}{}_{std} \times atwtK_{std}}{\omega_K \times f_{std}} \right]. \qquad (5)$$

Simplifying Eq. (5) yields



$$t_{std} = \frac{1}{\lambda_\lambda} ln \left[ 1 + \frac{\lambda}{\lambda_e} \frac{{}^{40}Ar^*_{std} \times atwtK_{std}}{\omega_K \times f_{std}} \right].$$ (6)

The age of a sample can be calculated using the ${}^{40}Ar/{}^{39}Ar$ age equation, updated with new λ, atwtK, and $f$ values, including $R$
= $F_{sample}/F_{standard}$ (Renne et al., 1998) where F = $({}^{40}Ar^*/{}^{39}Ar_K)$. We also define $r = f_{sample}/f_{standard}$ to account for the difference in $f$ $({}^{40}K/K)$ between samples and neutron fluence monitors:

$$t_{sam} = \frac{1}{\lambda_\lambda} ln \left[ 1 + R\, r\, (e^{\lambda_\lambda t_{std}} - 1) \right].$$ (7)

## 4 Effects of potassium isotope variability

Although $f$ values have not been measured on materials used to determine decay constants (e.g. Beckinsale and Gale, 1969) and those for most samples are not commonly measured directly, we can use values for silicate materials (in some cases directly relevant to ${}^{40}Ar/{}^{39}Ar$ geochronology) and evaporite materials to estimate the likely effects of potassium isotope variability on ${}^{40}Ar/{}^{39}Ar$ ages. One method for calculating the ${}^{40}Ar/{}^{39}Ar$ age of Fish Canyon sanidine (FCs) is based on an intercalibration with primary neutron fluence monitor GA1550 (Renne et al., 1998). Reasonable $f$ values are assumed, based on Morgan et al. (2018):

$f_\lambda$ based on the mean and standard deviation of evaporite values of $\delta^{41}K = 0.07 \pm 0.69‰$

$f_{std}$ based on the Mt. Dromedary biotite (GA1550) value of $\delta^{41}K = -0.77 \pm 0.12‰$

$f_{sample}$ based on the Fish Canyon sanidine (FCs) value of $\delta^{41}K = -0.51 \pm 0.16‰$.

Based on the above assumptions, the most likely scenario is that the K-Ar age of GA1550 is older than previously believed by ca. 35 ka, and the ${}^{40}Ar/{}^{39}Ar$ age of FCs (based on the age of GA1550) is older than previously believed by ca. 7 ka. Although these effects are small, they are within range of existing uncertainties in the ${}^{40}Ar/{}^{39}Ar$ system and should now be considered. Uncertainties in $\delta^{41}K$ will ultimately require propagation into the ${}^{40}Ar/{}^{39}Ar$ age equation, as part of continuing work on refining the ${}^{40}K$ decay constant, as well as ${}^{40}Ar$ and ${}^{40}K$ abundances in neutron fluence monitors.

Of course, most unknown sample materials used for ${}^{40}Ar/{}^{39}Ar$ geochronology are not FCs and may have a range of potential $f_{sample}$ values. Given the range of $\delta^{41}K$ values for silicates measured by Morgan et al. (2018), we can allow $f_{sample}$ to vary between values for $\delta^{41}K$ ranging from -1.5‰ to +1.5‰. Figure 2 shows the resulting 'sample' age for FCs over this range of $\delta^{41}K_{sample}$ values (left axis), the difference in sample age from the 'most likely scenario' described above (outside right axis), and the



relative effect on sample age (inside right axis). Variance over the range of most silicates, with $\delta^{41}K$ of -1 to 0‰, yields an

effect on sample ages (for this example, sample of ca. 28 Ma) of ca. 0.5‰.

The potential for variability in the $\delta^{41}K$ value of the material on which decay constants are measured can also be considered. Figure 3 shows the effects of $\delta^{41}K$ variability in both the sample and the decay constant material. Sample variability is discussed above, but variability in $\delta^{41}K$ of decay constant materials is unknown for published measurements (e.g. Beckinsale and Gale,

1969; Kossert and Günther, 2004; Malonda and Carles, 2002; Steiger and Jäger, 1977). Because all these measurements have been made on K-rich salts such as KCl and $KNO_3$, assuming that $\delta^{41}K$ of evaporites can represent likely $\delta^{41}K$ values for these materials is reasonable. Measuring the $^{40}K$ decay constants explicitly includes $\delta^{41}K$ measurements of the relevant materials.

**5 Conclusions**

$^{40}Ar/^{39}Ar$ geochronology currently assumes that $^{40}K/K$ values do not vary between samples, neutron fluence monitors, or

evaporites on which activity counting was done. Recent high-precision K isotope work has shown this to be a false assumption at the per-mil level. Quantifying the likely effects of potential $^{40}K/K$ variability yields a 'most likely' scenario where the age of GA1550 is found to be older by ca. 35 ka, and Fish Canyon sanidine is older by ca. 7 ka. Modeling of potential K isotope variability in samples, as well as standards and evaporites, yields a maximum age effect approaching 1.5‰ for extreme cases, but most silicates will be affected at sub-per-mil levels. Many remaining issues can be addressed by continuing work including

K isotope measurements, activity counting, and first principles measurements of $^{40}K$ and $^{40}Ar$ in common neutron fluence monitors.

**Acknowledgements**

I thank Paul Renne for discussions, and Bob Fleck and Ryan McAleer for their reviews. Any use of trade, product, or firm names is for descriptive purposes only and does not imply endorsement by the U.S. Government.

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



**Figure 1: δ⁴¹K values for terrestrial samples and standards, relative to NIST SRM 999b. Data are provided in Morgan et al. (2018). Uncertainties are estimated at the 2σ level and represent either within-run uncertainties (for samples with a single run) or precision between runs (for samples with multiple runs). Figure modified from Morgan et al. (2018). Arrows indicate analyzed samples important for geochronology.**






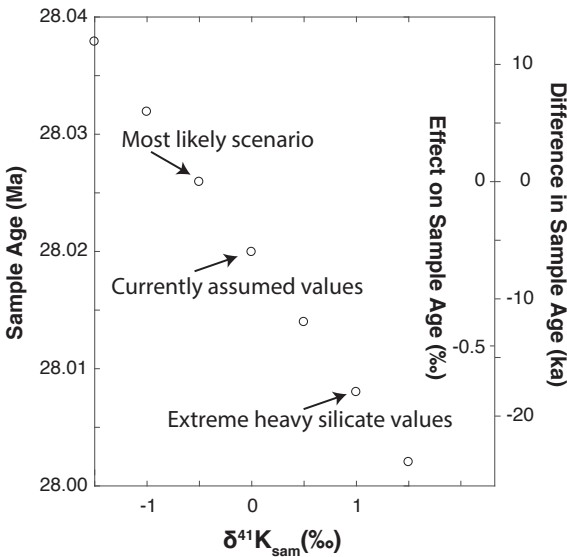


**Figure 2: Effects of varying the $\delta^{41}$K value of samples that are ca. 28 Ma. The left axis indicates the calculated sample age, whereas the right axis shows both the difference between the sample age and the 'most likely scenario' (outside right axis), and the per-mil effect on sample age. The arrows indicate the most likely scenario, the currently assumed values, and the most extreme heavy silicate values as measured by Morgan et al. (2018).**



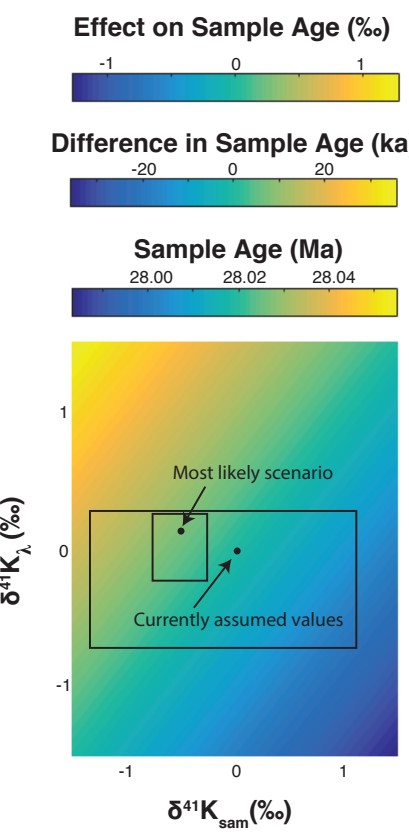


**Figure 3: Effects of varying the $\delta^{41}$K values of both the sample ($\delta^{41}$K$_{sam}$) and materials used to measure the decay constant ($\delta^{41}$K$_\lambda$). In this case, the sample is ca. 28 Ma. Indicated on the plot are currently assumed values, the most likely scenario as presented here, and two boxes indicating ranges of values. The smaller box represents the range of most silicate and evaporite materials, whereas the larger box includes the extremes of both groups, as measured by Morgan et al. (2018). As described in the text, the difference between currently assumed values and the most likely scenario results in age increases for GA1550 of +35 ka and FCs of +7 ka.**