# Peer review of "Potassium isotopic variability and implications for 40K-based geochronology"

_Geochronology, 2020_

## Referee Comment (RC1) · Ryan Ickert (Referee) · 15 Jul 2020

**General Comments**

This manuscript is a timely contribution to the literature – interest in K isotope variability is expanding, and the $^{40}$Ar/$^{39}$Ar community is increasingly pushing for improved traceability and precision. Potassium isotope variability has been long neglected – and with good reason – but this manuscript represents a timely contribution to the literature and is a good fit in *Geochronology*.

As I detail below, it is disappointing that the result in the manuscript is based on using one of the lesser used calibrations of $^{40}$Ar/$^{39}$Ar geochronology and note that this one result (which is the only result in this manuscript) has already been published – albeit

in shorter form – by Morgan et al. (2018). Fortunately, it should be straightforward for the effects of K isotope variability on the more relevant calibrations to be presented in a revised manuscript, so I don't think this is a serious problem.

The manuscript is not clear on how crucial quantities are calculated – in particular how $\delta$ $^{41}$K are converted to $^{41}$K/K or $^{41}$K/$^{39}$K. This problem is detailed below. These quantities are not tabulated, nor are the most other quantities in the manuscript. This makes the current draft difficult to read and hard to reproduce but it is an easy fix.

I hope to see a revised draft of this manuscript published in *Geochronology* and thank the author for bringing this issue to the attention of the geochronological community.

**Specific Comments**

**Why is only one type of $^{40}$Ar/$^{39}$Ar calibration described?**

As clearly stated in the manuscript, flux monitor calibration via a primary K-Ar reference material is only one way that that the monitor can be calibrated. This provides the most straightforward and most easily metrologically traceable calibration, but it is probably the least popular calibration at the present for high precision $^{40}$Ar/$^{39}$Ar measurements and hasn't been for quite some time. Both astronomical calibrations (e.g., Kuiper et al., Rivera et al.) and U-Pb calibrations (Renne et al. 2010/11) offer higher precision calibrations and are far more popular for high precision $^{40}$Ar/$^{39}$Ar work.

The author of the manuscript is obviously aware of this(!), but it's not clear why the effect of variable $^{40}$K/K are not also calculated for the far more popular and relevant calibrations. The few labs that use a primary calibration of GA1550 are probably not approaching limits of precision defined by K isotope variability and are unlikely to be interested in this result.

It's unclear to me why the more popular calibrations haven't been used here. If the resulting variation is tiny, that's also an important result.

I should also point out that this result – the effect of K isotope variability on GA1550 calibrated FCs – was already published by Morgan et al. (2018) in the top paragraph of page 185 of that paper. This result seems to be the main conclusion of the current manuscript and it's not clear that the relatively minor addition of the calculations is a significant advancement on the previously published result, by the same author.

I strongly recommend that

1. The effect of variable $^{40}$K/K is calculated for one or both of astronomically calibrated and U-Pb calibrated $^{40}$Ar/$^{39}$Ar.

**How are $^{40}$K/K and atomic weights derived?**

The manuscript is rather unclear as to how it calculates the $^{40}$K relative isotopic abundance. The manuscript uses measurements of the $^{41}$K/$^{39}$K, relative to SRM 999b ($\delta$ $^{41}$K). Via the measurements in Morgan et al. (2018), these can then be traced to SRM 985, which have a primary gravimetric calibration published by Garner et al. (1975). The $^{40}$K abundance (denoted $^{40}$K/K in the manuscript) is equal to the molar $^{40}$K/($^{39}$K+$^{40}$K+$^{41}$K), and by dividing by $^{40}$K is related to the isotope ratios via $^{40}$K/K = 1/(1+($^{39}$K/$^{40}$K)+($^{41}$K/$^{40}$K)). In the absence of new $^{40}$K measurements (relative to $^{39}$K or $^{41}$K), one way (perhaps the only way?) to obtain an estimate of $^{40}$K/K is to use the Garner et al. (1975) estimate for $^{41}$K/$^{39}$K of SRM 985, and through the delta-experiment traceability chain, calculate the $^{41}$K/$^{39}$K for the material of interest, assume a particular fractionation model (e.g., power law, exponential law etc.), and then calculate the $^{40}$K/$^{39}$K (or equivalently, the $^{40}$K/$^{41}$K) from the degree of fractionation measured by the delta-experiments and the fractionation law.

The manuscript employs ratios of $^{40}$K/K (e.g., $^{40}$K/K$^{garner}$/$^{40}$K/K$^{new}$) and constructs equations for decay constants and ages for flux monitors and standards using this ratio (denoted "r" in the paper). It might be assumed that the manuscript calculates

$^{40}$K/K, but it is not made clear. What I suspect that has been done is that they have assumed that the magnitude of the fractionation of $^{40}$K/$^{39}$K (hopefully relative to SRM 985 and not SRM 999b?) is half of what has been measured for $^{41}$K/$^{39}$K – effectively assuming a linear fractionation law (which is not an appropriate fractionation law, but is probably close enough). The manuscript then probably also makes the assumption that the magnitude of the fractionation in delta space is the same as the fractionation in "fraction" space, e.g.,

($^{40}$K/$^{39}$Kmeasured/$^{40}$K/$^{39}$Kgarner – 1) = ($^{40}$K/Kmeasured/$^{40}$K/Kgarner – 1)

This equality is approximately true if $^{40}$K/$^{39}$K and $^{40}$K/K are very small, but it should be noted in the manuscript that this is not exactly correct and is more significantly wrong for larger ratios such as $^{41}$K/$^{39}$K (where a 1 ‰ in delta space is 0.067 ‰ in fraction space).

A similar lack of clarity applies to the calculation of the atomic weight. An atomic weight requires knowledge of the relative isotope abundances of all the isotopes in a sample and will change if the isotope composition changes. Potassium-40 has a very small influence on the atomic weight (it adds $\sim$ 120 ppm) so changes in its relative abundance can be ignored (as was done in Morgan et al., (2018)), but each distinct isotope composition used should have its own atomic weight. The calculation is not described, nor are the calculated quantities.

I understand that some of the effort expended to use $^{40}$K/K "ratio ratios" was so that the decay constants can be adjusted from published values, and that they do not need to be calculated anew. This is quite clever, and eminently reasonable. But the full K isotope compositions used should be written in the text or tabulated so that the calculations can be reproduced. With all the sleuthing necessary to work out what has been done, and apparent shortcuts taken via approximations, it is extraordinarily difficult to reproduce the calculations because the quantities of interest are never tabulated. Tabulating the results is straightforward, because we know what the Garner et al. (1975) $^{40}$K/K value

is (0.01167), and the manuscript at some point has calculated the difference between the $^{40}$K/K of a sample and the $^{40}$K/K of SRM 999b (which is traceable to SRM 985, from which the Garner et al. quantities are derived).

So to summarize, the manuscript would benefit from:

1. Detailing quantitatively, using equations, how to go from $\delta$ $^{41}$K (relative to SRM 999b) to $^{40}$K/K, or even better, $^{41}$K/$^{39}$K and $^{40}$K/$^{39}$K) and how to calculate the atomic weight.

2. Tabulating the $^{40}$K/K (or even better, $^{41}$K/$^{39}$K and $^{40}$K/$^{39}$K) and atomic weights used in the calculations.

**Generalizability of results**

The manuscript is quite short, which is admirable, but since it's only applied to a single calibration, and a sample of a single age, it's difficult for readers to know how this might apply to samples of other ages. It would be useful to include a wider range of sample ages (alongside other calibrations).

The results (e.g., figure 2) are couched in terms of "bias" , considering extreme values. In my opinion, it would be far more useful if the distribution of $^{40}$K/K was calculated, and a probabilistic result was presented. In other words, treating this as an uncertainty in the $^{40}$K/K and propagating that uncertainty onto the age results. A very close analogy is the effect of variable $^{238}$U/$^{235}$U in U-Pb geochronology, and Fig 2 from Hiess et al. (2012; Science v335 p1610), where the bias as a function of time is plotted, with a band representing the variability in 238/235.

I appreciate that this is extra work for the author, but a straightforward MonteCarlo error propagation can easily be done in most software packages (even excel) in a few minutes – there is no need to derive cumbersome error propagation equations. This

would expand the generality of this manuscript enormously and benefit anyone not dating something that is 28 Ma.

The magnitude of the variability in age associated with $^{40}$K/K variability should be compared to other systematic uncertainties. Is the decay constant uncertainty large or smaller? Is flux monitor age uncertainty larger or smaller? This will give non-specialists a sense of whether this is a first or second order problem to tackle.

**Technical Corrections**

The (approximate) isotope composition of K should be described in the introduction for a reader who is not intimately familiar with isotopes of K.

The Merrihue and Turner reference in the first paragraph is confusing because the placement implies that this is the reference for the half life, and not for $^{40}$Ar/$^{39}$Ar geochronology. It might be worthwhile substituting a more general reference, like Harrison and McDougal, that encompasses both.

I think that at this point in geochronology Steiger and Jager is not really a good reference for decay rates and branching ratios. It describes an agreed "convention" and doesn't detail how the quantities were compiled or derived. Beckinsale and Gale or Min et al. are probably more appropriate. Or Renne et al. (2010/2011) if you really want to stir the pot.

The notation of $^{40}$K/K is peculiar and possibly confusing. $^{40}$K/K here means the molar $^{40}$K/($^{39}$K+$^{40}$K+$^{41}$K) and is not really a standard notation in isotope geochemistry. It's not obvious to me what a suitable difference would be – my preference would be to simply write $^{40}$K/$^{39}$K in the text, and then if necessary, use $^{40}$K/K in equations (or another symbol indicating fractional $^{40}$K). In any case, $^{40}$K/K needs to be defined quantitatively somewhere in the manuscript.

Other notations are also confusing. The use of lower case "f" and "r" when the uppercase letters are in common use for unrelated quantities in the equations makes this very difficult to read – the sentence on line 80 is a good example of this. It would be more straightforward if Greek letters were substituted.

All quantities used in calculations, decay constants, atomic weights, ages, and other constants should be either listed in the text or tabulated. Almost none of them are listed in the text. They should all be explicitly referenced.

In equation six the symbol for the decay constant is lambda with a subscript lambda. I assume that this is a typo and that it is meant to be lambda subscript new.

Figure 1 is only very slightly modified from Figure 6 in Morgan et al. (2018), which may be a copyright violation. I appreciate that the lead author of that paper is the author of this submission, and that there are only so many ways in which a ranked-data plot can be drafted, and I am not accusing the author of plagiarism or an ethical violation. However, it's clear that the same electronic figure was used in this submission – the fonts, colors, spacing, and all the style characteristics are identical. The "modifications" appear to be just a few extra lines and a couple of arrows. I would urge one of two actions, either the figure be substantially modified so that it no longer resembles that in Morgan et al. (2018), or the publication staff of *Geochronology* confirm that they can legally print this figure via an existing license from the RSC (the publisher of JAAS) or a one-off agreement.

Figure 2: The dots are presumably point estimates of a continuous function, so the curve should be drawn instead of the points, and the plausible range should be bracketed. The point estimates outside of the range were confusing initially before I realized there was an implied curve denoting a continuous function. As mentioned above, this figure would benefit from redrafting as a "change in age vs. absolute age" with an uncertainty band derived from different values of $^{40}$K/K.

Figure 3: The colours are nice, but the figure would be easier to read if it were made wider and had a few labelled contours instead of colours. One has to look back and

forth from the colours to see what the quantities are, and the contours will just be straight lines. They could easily be labelled without cluttering the chart if it were wider, and they could be labelled outside the top and right edges of the figure.

---

## Referee Comment (RC2) · Anonymous Referee #2 · 17 Jul 2020

General Comments This manuscript is a welcome contribution to the literature. With technical and scientific advancements in the Ar/Ar community, interest in sources of uncertainty once thought to be second order has expanded greatly. Potassium isotope variability has long been neglected and it's important to investigate its effect on the Ar/Ar system, however, even the author admits the effects appear to be minor.

Overall I admire the concise technical manuscript focused on pushing forward our understanding of the systematic uncertainties in Ar/Ar geochronology. With minor revisions, as detailed below, I hope to see this manuscript published in Geochronology.

Specific Comments Comparison of the systematic uncertainties. The manuscript does a nice job of demonstrating the effect of Potassium isotope variability. However the magnitude of this systematic error is not put into any context. I strongly recommend

[Figure]

a comparison and discussion of the other major systematic uncertainties. When does the community need to consider this issue? What are examples where this systematic uncertainty has implications for open geoscience questions? How does this compare to other minor but consequential sources of systematic uncertainty, such as "cold storage", radial and circumferential flux ratio, mass spectrometer biases such as mass discrimination and detector intercalibration?

Should age of FC be revised? "Based on the above assumptions, the most likely scenario is that the K-Ar age of GA1550 is older than previously believed by 95 ca. 35 ka, and the 40Ar/39Ar age of FCs (based on the age of GA1550) is older than previously believed by ca. 7 ka." Is the author suggesting a revised age for GA1550 and FCs? What additional work is required to reinforce this result?

Comparison to more neutron flux standards The manuscript compares the effect of k isotope variability to only GA1550 and FC. Although these are widely used neutron flux monitors other monitors are routinely used and could be mentioned, namely ACs. The community is also in the process of identifying new potential neutron flux monitors and more discussion and examples of how this systematic uncertainty can affect interlaboratory and inter-system calibration is important.

Technical Comments I recommend improving the nomenclature and symbols used in the equations. I was able to follow the calculations but at times found it difficult. I recommend changing atwtK, to AK or similar.

I recommend improving figure 2. The effect on age is a continuous function and should be represented as such. Where the major neutron flux monitors plot should be included, not simply "currently assumed values". A vertical shaded region indicated the typical and extreme ranges of delta 41K for silicates is recommended.

I also recommend adding another figure showing the effect on sample age at different age ranges, perhaps at 1,10,100 Ma.

Line 112 "Measuring the 40K decay constants explicitly includes d41K measurements of the relevant materials" This statement needs more context or somehow incorporated into the paragraph better. Its not clear the purpose of this statement in the context of the preceding paragraph.

———————————————————

---

## Author Comment (AC1) · 29 Aug 2020

I would like to thank Ryan Ickert for his thoughtful and thorough review. We will respond to his comments here.

Comment: I strongly recommend that 1. The effect of variable 40K/K is calculated for one or both of astronomically calibrated and U-Pb calibrated 40Ar/39Ar.

Response: We agree that adding these calibrations would make the manuscript more useful. This is relatively simple for the astronomical calibration, and this will be added to a revised manuscript.

Comment: So to summarize, the manuscript would benefit from:

[Figure]

1. Detailing quantitatively, using equations, how to go from $\delta 41K$ (relative to SRM 999b) to 40K/K, or even better, 41K/39K and 40K/39K) and how to calculate the atomic weight.

Response: These equations will be presented in a revised manuscript.

2. Tabulating the 40K/K (or even better, 41K/39K and 40K/39K) and atomic weights used in the calculations.

Response: Parameters used will be tabulated in a revised manuscript.

Comment: The results (e.g., figure 2) are couched in terms of "bias", considering extreme values. In my opinion, it would be far more useful if the distribution of 40K/K was calculated, and a probabilistic result was presented. In other words, treating this as an uncertainty in the 40K/K and propagating that uncertainty onto the age results... The magnitude of the variability in age associated with 40K/K variability should be compared to other systematic uncertainties. Is the decay constant uncertainty large or smaller? Is flux monitor age uncertainty larger or smaller?

Response: Uncertainty in 40K/K will be propagated into age equations in a revised manuscript.

Technical Corrections:

The (approximate) isotope composition of K should be described in the introduction for a reader who is not intimately familiar with isotopes of K.

Response: This will be added to a revised manuscript.

The Merrihue and Turner reference in the first paragraph is confusing because the placement implies that this is the reference for the half life, and not for 40Ar/39Ar geochronology. It might be worthwhile substituting a more general reference, like Harrison and McDougal, that encompasses both.

Response: This sentence will be rearranged and will include the McDougall and Harri-

son reference.

I think that at this point in geochronology Steiger and Jager is not really a good reference for decay rates and branching ratios. It describes an agreed "convention" and doesn't detail how the quantities were compiled or derived. Beckinsale and Gale or Min et al. are probably more appropriate. Or Renne et al. (2010/2011) if you really want to stir the pot.

Response: Beckinsale and Gale, as well as Min et al., will be added as references here.

The notation of 40K/K is peculiar and possibly confusing. 40K/K here means the molar 40K/(39K+40K+41K) and is not really a standard notation in isotope geochemistry. It's not obvious to me what a suitable difference would be – my preference would be to simply write 40K/39K in the text, and then if necessary, use 40K/K in equations (or another symbol indicating fractional 40K). In any case, 40K/K needs to be defined quantitatively somewhere in the manuscript.

Response: 40K/K is used in literature going back at least to Beckinsale and Gale, and including Min et al. 40K/39K would not be accurate as it doesn't include 41K in the denominator. 40K/K will be defined in section 1, and 'f' will be retained in equations for consistency with symbols used by Min et al.

Other notations are also confusing. The use of lower case "f" and "r" when the upper case letters are in common use for unrelated quantities in the equations makes this very difficult to read – the sentence on line 80 is a good example of this. It would be more straightforward if Greek letters were substituted.

Response: As noted above, the use of 'f' is for consistency with Min et al. 'r' was selected as the ratio of 'f/f' for symmetry: r=f/f, compared with the equation R=F/F.

All quantities used in calculations, decay constants, atomic weights, ages, and other constants should be either listed in the text or tabulated. Almost none of them are listed

GChronD

in the text. They should all be explicitly referenced.

Response: These will be tabulated in a revised manuscript.

In equation six the symbol for the decay constant is lambda with a subscript lambda. I assume that this is a typo and that it is meant to be lambda subscript new.

Response: This is actually not a typo. ïĄňïĄň, while a slightly awkward symbol, represents the decay constant calculated using ïĄď41K values for material used in activity counting. This is defined just below equation 4.

Figure 1 is only very slightly modified from Figure 6 in Morgan et al. (2018), which may be a copyright violation. I appreciate that the lead author of that paper is the author of this submission, and that there are only so many ways in which a ranked-data plot can be drafted, and I am not accusing the author of plagiarism or an ethical violation. However, it's clear that the same electronic figure was used in this submission – the fonts, colors, spacing, and all the style characteristics are identical. The "modifications" appear to be just a few extra lines and a couple of arrows. I would urge one of two actions, either the figure be substantially modified so that it no longer resembles that in Morgan et al. (2018), or the publication staff of Geochronology confirm that they can legally print this figure via an existing license from the RSC (the publisher of JAAS) or a one-off agreement.

Response: The RSC does not require authors to obtain permission to reproduce material from their own works (see quote below). The appropriate permission will be listed in the figure caption.

From RSC: "If you are the author of this article you do not need to formally request permission to reproduce figures, diagrams etc. contained in this article in third party publications or in a thesis or dissertation provided that the correct acknowledgement is given with the reproduced material."

Figure 2: The dots are presumably point estimates of a continuous function, so the

curve should be drawn instead of the points, and the plausible range should be bracketed. The point estimates outside of the range were confusing initially before I realized there was an implied curve denoting a continuous function. As mentioned above, this figure would benefit from redrafting as a "change in age vs. absolute age" with an uncertainty band derived from different values of 40K/K.

Response: The curve will be redrawn as a continuous function, and plausible ranges noted.

Figure 3: The colours are nice, but the figure would be easier to read if it were made wider and had a few labelled contours instead of colours. One has to look back and forth from the colours to see what the quantities are, and the contours will just be straight lines. They could easily be labelled without cluttering the chart if it were wider, and they could be labelled outside the top and right edges of the figure.

Response: This figure will be redrawn with contours in a revised manuscript.

―――――――――――――――――――――

---

## Author Comment (AC2) · 29 Aug 2020

Thanks to this anonymous reviewer for their positive and constructive comments, which are responded to here.

Comment: I strongly recommend a comparison and discussion of the other major systematic uncertainties. When does the community need to consider this issue? What are examples where this systematic uncertainty has implications for open geoscience questions? How does this compare to other minor but consequential sources of systematic uncertainty, such as "cold storage", radial and circumferential flux ratio, mass spectrometer biases such as mass discrimination and detector intercalibration?

Response: These are important questions that are difficult to answer. Ultimately, the

other sources listed will vary considerably with sample and time, and so will K isotopic variability. A statement will be added to the conclusions indicating that this is a smaller issue than many uncertainty sources but may become more important as our constraints on those uncertainties improve.

Comment: Is the author suggesting a revised age for GA1550 and FCs? What additional work is required to reinforce this result?

Response: Given the relatively small effect on these ages, and ongoing work on this issue, we are not currently suggesting revised ages for neutron fluence monitors. Rather, we are publishing these results to make this issue known and providing the equations so that it can be accounted for when necessary. Ultimately, renewed work measuring 40K concentrations in mineral standards will likely result in revised ages, but that work is yet to be completed.

Comment: . . . other monitors are routinely used and could be mentioned, namely ACs. The community is also in the process of identifying new potential neutron flux monitors and more discussion and examples of how this systematic uncertainty can affect interlaboratory and inter-system calibration is important.

Response: This is a good point. The effect on the age of ACs will be modeled and shown in the revised manuscript.

Technical Comments:

Comment: I recommend improving the nomenclature and symbols used in the equations. I was able to follow the calculations but at times found it difficult. I recommend changing atwtK, to AK or similar.

Response: In the revised manuscript, 'atwtK' will be changed to 'W', for consistency with Min et al. (2000).

Comment: I recommend improving figure 2. The effect on age is a continuous function and should be represented as such. Where the major neutron flux monitors plot should

be included, not simply "currently assumed values". A vertical shaded region indicated the typical and extreme ranges of delta 41K for silicates is recommended.

Response: Figure 2 will be improved as suggested.

I also recommend adding another figure showing the effect on sample age at different age ranges, perhaps at 1,10,100 Ma.

Response: We agree this would be useful and will work to develop such a figure for the revised manuscript.

Line 112 "Measuring the 40K decay constants explicitly includes d41K measurements of the relevant materials" This statement needs more context or somehow incorporated into the paragraph better. Its not clear the purpose of this statement in the context of the preceding paragraph.

Response: This sentence will be reworded for clarity: "Future measurements of 40K decay constants should include ïĄď41K measurements of the relevant materials."